# Generative Pre-trained Autoregressive Diffusion Transformer

**Yuan Zhang**[1*]**, Jiacheng Jiang**[2*]**, Guoqing Ma**[3*†]**, Zhiying Lu**[4]**, Haoyang Huang**[3]**, Jianlong Yuan**[3]

Nan Duan[3‡]

[1]Peking University
[2]Tsinghua University [3]StepFun, China
[4]University of Science and Technology of China

## Abstract

In this work, we present GPDiT, a Generative Pre-trained Autoregressive Diffusion Transformer that unifies the strengths of diffusion and autoregressive modeling for long-range video synthesis, within a continuous latent space. Instead of predicting discrete tokens, GPDiT autoregressively predicts future latent frames using a diffusion loss, enabling natural modeling of motion dynamics and semantic consistency across frames. This continuous autoregressive framework not only enhances generation quality but also endows the model with representation capabilities. Additionally, we introduce a lightweight causal attention variant and a parameter-free rotation-based time-conditioning mechanism, improving both the training and inference efficiency. Extensive experiments demonstrate that GPDiT achieves strong performance in video generation quality, video representation ability, and few-shot learning tasks, highlighting its potential as an effective framework for video modeling in continuous space.

## 1 Introduction

Diffusion models have achieved notable success in video generation [1, 2, 11, 14]. Despite recent advancements, existing diffusion-based approaches exhibit limitations in temporal consistency and motion coherence, particularly in long-range generation. A key contributing factor is the use of bidirectional attention. This allows future context to influence current predictions, thereby violating the causal structure required for autoregressive generation.

In contrast, autoregressive (AR) modeling has become the de facto paradigm in natural language processing [3, 32, 33]. This method inherently captures the causality in sequences by predicting the next token based on previously generated outputs, thereby facilitating both sequence modeling and structural understanding. Inspired by this, recent efforts [7, 9, 48] have focused on integrating AR modeling with diffusion processes to leverage their complementary strengths. This integration aims to improve temporal coherence and enhance motion continuity in extended video synthesis. Notably, compared to traditional cross-entropy objectives that explicitly train for next-token prediction, the combination of diffusion-based loss and AR modeling offers an implicit path to temporal understanding, serving as a natural byproduct rather than a specially designed objective.

One line of research investigates the replacement of bidirectional attention with causal attention [4, 8, 10, 16, 56] for improved modeling of temporal dependencies. Specifically, the attention computation restricts each noisy token to attend only to preceding clean tokens and itself. This

---

[*]Equal contribution.

[†]Technical leader.

[‡]Corresponding author.

39th Conference on Neural Information Processing Systems (NeurIPS 2025).

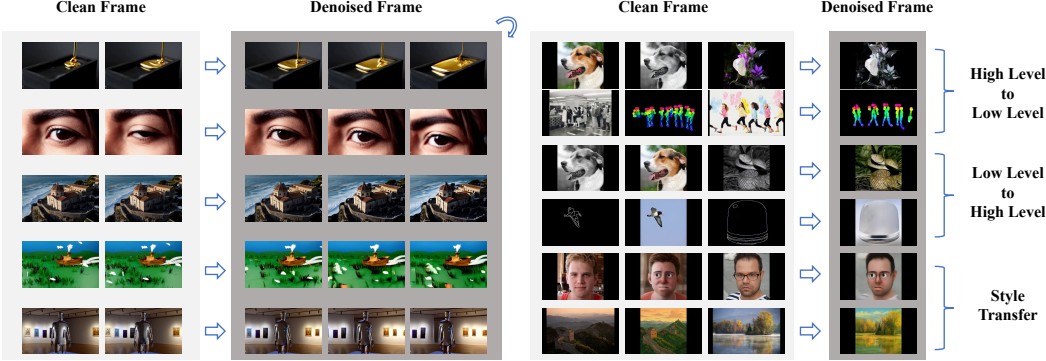

**Figure 1:** Video Generation and Few-Shot Multitask Learning. The left side of the figure illustrates the model's video generation capability: given a set of initial frames, the model can continue the sequence by generating denoised frames. The right side showcases the model's multitask learning ability, similar to the approach presented in [33]. After few-shot fine-tuning, the model is capable of performing a variety of tasks, such as translating high-level features to low-level features, converting low-level features to high-level features, and executing style transfer across video sequences.

architectural design enables more robust generalization to video lengths beyond those seen during training, whereas bidirectional attention often leads to severe quality degradation when extrapolating to longer sequences. Moreover, causal attention is naturally compatible with KV cache, significantly accelerating the generation of long sequences. These advantages are unattainable with traditional bidirectional attention mechanisms.

Another increasingly studied approach is diffusion forcing [5, 19, 37, 40], characterized by the asynchronous injection of token-specific noise levels during training to facilitate long-range dependency modeling. During inference, this method operates in a coarse-to-fine manner: it first generates early frames and then progressively refines later ones through iterative denoising. While promising, this paradigm still struggles with training instability, with independent frame-level noise schedules often impairing performance compared to synchronized alternatives.

To address the aforementioned challenges in long-sequence video modeling, we introduce Generative Pre-trained Autoregressive Diffusion Transformer (GPDiT), a frame-wise autoregressive diffusion framework. In contrast to discrete token-level autoregressive modeling, GPDiT captures causal dependencies across frames while preserving full attention within each frame, enabling both sequential coherence and intra-frame expressivity. Moreover, GPDiT improves training and inference efficiency through two practical architectural modifications. The first component introduces an attention mechanism that leverages the temporal redundancy of video sequences by eliminating attention computation between clean frames during training, thereby reducing computational cost without compromising generation performance. The second is a parameter-free time-conditioning strategy that reinterprets the noise injection process as a rotation in the complex plane defined by data and noise components. This design removes the need for adaLN-Zero [30] and its associated parameters, yet still effectively encodes time information. As shown in Figure 1, GPDiT performs well in both video generation and few-shot learning tasks.

Our main contributions can be summarized as follows:

- We introduce GPDiT, a strong autoregressive video generation framework that leverages framewise causal attention to improve temporal consistency over long durations. To further enhance efficiency, we propose a lightweight variant of causal attention that significantly reduces computational costs during both training and inference.

- By reinterpreting the forward process of diffusion models, we introduce a rotation-based conditioning strategy, offering a parameter-free approach to inject time information. This

lightweight design eliminates the parameters associated with adaLN-Zero while achieving model performance on par with state-of-the-art DiT-based methods.

- Extensive experiments demonstrate that GPDiT achieves competitive performance on video generation benchmarks. Furthermore, evaluations on video representation tasks and few-shot learning tasks show its potential of video understanding capabilities.

## 2 Related works

**Video Diffusion models.** Diffusion and flow-based generative models [13, 22, 23, 38, 43, 50] have demonstrated unprecedented ability to capture visual concepts and produce high-quality images. Video Diffusion Models [14] is the first work to introduce diffusion models for video generation. However, the expense associated with pixel space diffusion and denoising is nontrivial, requiring substantial computational resources. Models such as Magicvideo [55] and LVDM [11] speed up training and sampling efficiency by compressing high-dimensional video data into a latent space. Recent efforts have scaled diffusion transformers to substantially larger capacities, demonstrating significantly enhanced generation capabilities and further revolutionizing the field of text-to-video (T2V) synthesis, driving rapid progress across a broad range of applications [15, 17, 21, 25, 26, 28**?** , 36, 41, 44, 45, 47**?** ].

**Autoregressive Modeling.** An emerging trend is the integration of autoregressive modeling and diffusion models. A key characteristic of this approach is that the model predicts future videos based on previously generated content. Representative works, such as [20**?** , 46, 52, 54], leverage an additional visual tokenizer that maps pixel-space inputs into discrete tokens, which are then fed into a language model to generate videos. However, this mapping is inherently lossy, leading to inferior performance compared to video diffusion models. Recent works [10, 24, 40, 56], inspired by Diffusion Forcing [5], allow each video frame to be processed with distinct noise levels, enabling the generation of videos with variable lengths. However, adding noise to antecedent sequences complicates future predictions and degrades performance on discriminative tasks. Recent work [**?** ] addresses this prediction ambiguity by preserving the clarity of antecedent representations, keeping them noise-free.

## 3 Preliminary

### 3.1 Denoising Diffusion

Diffusion models generate data by progressively corrupting samples from the data distribution with Gaussian noise, eventually transforming them into pure noise, and then learning to invert this process through a sequence of denoising steps. Formally, given a data distribution $p_{\text{data}}(x)$, diffusion models apply a stochastic differential equation (SDE) to gradually perturb the data:

$$dx_t = \mu(x_t, t)\, dt + \sigma(t)\, dw_t, \tag{1}$$

where $t \in [0, T]$ for some fixed terminal time $T > 0$, $\mu(\cdot, \cdot)$ denotes the drift coefficient, $\sigma(\cdot)$ is the diffusion coefficient, and $\{w_t\}_{t \in [0,T]}$ represents a standard Brownian motion. Let $p_t(x)$ denote the marginal distribution of $x_t$; by construction, the initial distribution satisfies $p_0(x) = p_{\text{data}}(x)$. A notable property of this SDE is the existence of a corresponding ordinary differential equation (ODE), referred to as the Probability Flow ODE [38], whose solution trajectories are guaranteed to match the time-evolving marginals $p_t(x)$:

$$dx_t = \left[ \mu(x_t, t) - \frac{1}{2}\sigma(t)^2 \nabla \log p_t(x_t) \right] dt. \tag{2}$$

## 4 Generative Pre-trained Autoregressive Diffusion Transformer (GPDiT)

In this section, we present an effective framework that combines autoregressive and diffusion models for video modeling. First, we introduce two variants of the attention mechanism tailored for frame-aware autoregressive diffusion in Section 4.1. Then, we discuss a flexible conditioning strategy designed to handle both clean and noisy frames in Section 4.2. Figure 2 provides an overview of the GPDiT framework, illustrating the inference pipeline, the internal architecture of a GPDiT block, and the rotation-based interpretation of the diffusion process.

## 4.1 Attention mechanism

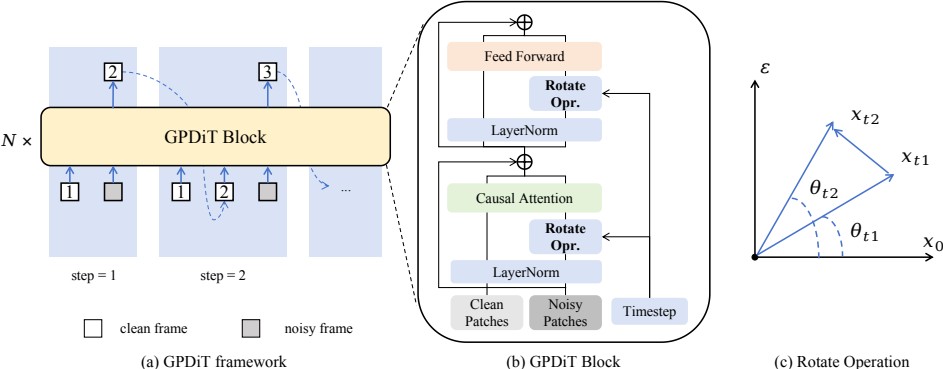

| (a) GPDiT framework | (b) GPDiT Block | (c) Rotate Operation |

Figure 2: *Left plane:* An overview of GPDiT inference. *Middle plane:* The architecture of a typical GPDiT block, where adaLN-Zero is replaced with our rotation-based time conditioning, and causal attention is adopted instead of conventional bidirectional attention. *Right plane:* An illustration of the rotation-based view of the diffusion forward process, where the data and noise components evolve through a parameter-free rotation in the complex plane.

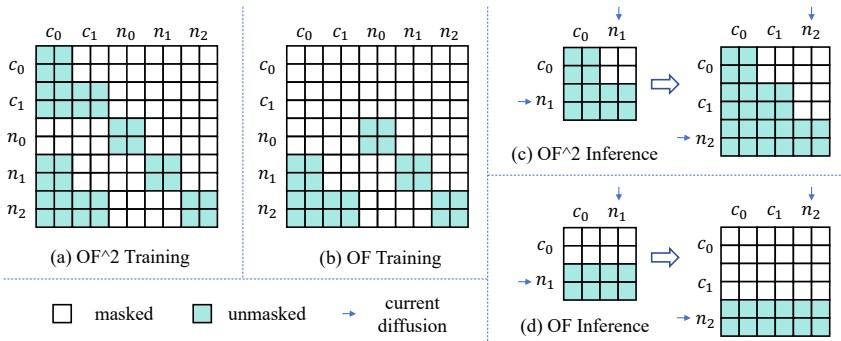

Figure 3: Illustration of two causal attention variants. Both apply intra-frame full attention and inter-frame causal attention, but differ in cross-frame attention handling between clean frames. $c_i$ and $n_i$ denote clean and noisy frames, respectively.

### 4.1.1 Vanilla Causal Attention

The traditional bidirectional attention mechanism has been criticized for disrupting temporal coherence and failing to maintain consistency in long video modeling. Meanwhile, most existing models struggle to produce high-quality videos that exceed the frame length they were trained on since the models can only learn the joint distribution of fixed-length frames. To alleviate the issues, we employ the standard causal attention shown in Figure 3 (a) and (c), where each noisy frame $n_i$ can only attends to previous clean frames $c_{<i}$ and itself, while $c_{<i}$ also attend to each other. The training objective is:

$$\mathcal{L}(\theta) = \mathbb{E}_{t \sim U[0,1], \epsilon \sim \mathcal{N}(0,I), x \sim p_{data}} \|\epsilon(n_i, t \mid c_{<i}) - \epsilon^t\|^2. \tag{3}$$

A notable advantage of standard causal attention is its compatibility with key-value (KV) caching [31] during inference, significantly accelerating generation and shortening the time needed for long video production.

### 4.1.2 Lightweight Causal Attention

Although the advantages of vanilla causal attention are notable, it presents two major challenges. First, during training, maintaining a clean copy of the noised sequence for attention map computation doubles the memory and computational costs. Second, during inference, the inevitable expansion of

the KV cache due to token accumulation in long-sequence prediction imposes a prohibitive memory burden.

To address these issues, we propose a lightweight causal attention mechanism that exploits the spatial redundancy in video data. As illustrated in Figures 3 (b) and (d), we eliminate attention score computation between clean frames, thus reducing additional operations without compromising model performance. To quantify computational savings, we analyze the attention complexity in the vanilla design. The computational overhead can be decomposed into three components: attention between clean contexts, attention between noisy frames and clean contexts, and self-attention among noisy frames, with computational complexities of $\mathcal{O}(\frac{1}{2}F^2)$, $\mathcal{O}(\frac{1}{2}F^2)$, and $\mathcal{O}(F)$, respectively, where $F$ denotes the number of frames. Since attention between clean frames accounts for nearly half of the total computation, its removal leads to a substantial reduction in training cost. Moreover, during inference, achieving $\mathcal{O}(F)$ complexity with standard causal attention requires maintaining a key-value (KV) cache, resulting in an additional $\mathcal{O}(2F)$ memory overhead. In contrast, our method attains $\mathcal{O}(F)$ inference complexity without incurring extra memory costs, substantially reducing the memory footprint.

## 4.2 Re-Thinking Timestep Conditioning Injection

The Adaptive Normalization Layer Zero (adaLN-Zero) has been widely utilized to incorporate timestep and class-label embeddings into diffusion model backbones, as introduced in DiT [30]. adaLN-Zero is typically designed as an MLP block to extract class-label embeddings for each transformer block. However, modern tasks in text-to-image, text-to-video, and image-to-video generation involve more complex semantic embeddings. These embeddings are often injected into the model through techniques such as token concatenation along the sequence dimension or cross-attention, leaving the MLP block to primarily handle timestep embeddings. The authors of [6] argue that the adaLN-Zero submodule contributes significantly to the model's parameter count, accounting for an increase of approximately 28%. This considerable overhead has motivated research into more efficient methods for incorporating time conditioning in these models, aiming to reduce the computational cost while maintaining or enhancing performance.

We begin by considering the forward (variance-preserving) diffusion process, given by:

$$x_t = \sqrt{\bar{\alpha}_t}x_0 + \sqrt{1 - \bar{\alpha}_t}\epsilon,$$

where $x_0 \in \mathbb{R}^D$ is a clean sample drawn from the data distribution, $\epsilon \sim \mathcal{N}(0, I)$ represents standard Gaussian noise, and $\alpha_t \in [0, 1]$. To facilitate our analysis, we reduce the problem to one dimension ($D = 1$) and reinterpret the forward process as a rotation in a 2D space. Specifically, we define the rotation angle $\theta_t$ as:

$$\cos\theta_t = \sqrt{\bar{\alpha}_t}, \quad \sin\theta_t = \sqrt{1 - \bar{\alpha}_t},$$

such that the forward process becomes:

$$x_t = \cos\theta_t x_0 + \sin\theta_t \epsilon.$$

To represent this process geometrically, we stack the clean sample $x_0$ and the noise $\epsilon$ into a 2-vector $\begin{pmatrix} x_0 \\ \epsilon \end{pmatrix} \in \mathbb{R}^2$. The forward diffusion step is then represented as an orthogonal rotation in this 2D space:

$$\begin{pmatrix} x_t^{(0)} \\ x_t^{(1)} \end{pmatrix} = \underbrace{\begin{pmatrix} \cos\theta_t & \sin\theta_t \\ -\sin\theta_t & \cos\theta_t \end{pmatrix}}_{R(\theta_t)} \begin{pmatrix} x_0 \\ \epsilon \end{pmatrix},$$

In this formulation, $x_t^{(0)}$ represents the usual diffused sample, while $x_t^{(1)}$ is its orthogonal complex companion. The clean sample $x_0$ and the noise $\epsilon$ can be recovered by applying the inverse rotation:

$$\begin{pmatrix} x_0 \\ \epsilon \end{pmatrix} = R(\theta_t)^{-1} \begin{pmatrix} x_t^{(0)} \\ x_t^{(1)} \end{pmatrix}.$$

The model is trained to predict the complex companion $x_t^{(1)}$ from the input $x_t^{(0)}$ using a predefined loss function, which is assumed to be unknown for the current analysis.

The proposed approach follows the principle of parsimony, applying a reverse rotation with the angle $\theta_t$ on $x_t^{(0)}$ for each block to efficiently inject the timestep embedding while incurring no additional computational overhead. Other forms of conditioning, such as text or image conditioning, can be incorporated in the standard manner.

# 5    Experiments

## 5.1    Experimental Setups

We conduct experiments in three scenarios: video generation, video representation, and few-shot learning. The results demonstrate that GPDiT exhibits excellent generative and representational capabilities, which are crucial for building a unified model for visual understanding and generation, as well as the ability to transfer to downstream tasks with minimal cost and no need for additional modules.

**Datasets.** For video generation task, UCF-101 [39] dataset consists of 13,320 videos across 101 action categories and is widely used for human action recognition, MSR-VTT [49] is a large-scale dataset designed for open-domain video captioning, containing 10,000 video clips from 20 categories, with each clip annotated with 20 English sentences by Amazon Mechanical Turk workers. We assess the capability of GPDiT in video representation on the UCF-101 dataset. For the few-shot learning tasks, we construct multiple supervised fine-tuning (SFT) datasets. For each task, we create a SFT dataset with 20 video sequences, each generated by sampling three pairs from a set of 40 task-specific image pairs. These tasks include human detection, image colorization, Canny edge-to-image reconstruction, and two style transfer applications.

**Evaluations.** For video generation, we randomly sample 10,000 videos from UCF-101 and 7,000 videos from MSR-VTT. The Fréchet Video Distance (FVD) [42] is computed for entire videos, while the average Fréchet Inception Distance (FID) [12] and Inception Score (IS) [34] are calculated over individual frames. For the video representation task, top-1 accuracy is reported using a linear probing protocol. In the few-shot learning setting, we provide per-task video results along with qualitative analyses.

**Implementation details.** To ensure fair comparison, we design a benchmark model with 80 million parameters based on the architecture in Table 1. Trained on UCF-101, each video is center-cropped and resized to 256×256. The Adam optimizer with a learning rate of 1e-4 and a total batch size of 96 across 32 H100 GPUs is used. Training lasts for 400k iterations.

Table 1: Model variants of GPDiT. We follow the model size configurations of DiT [30] and SiT [27], replacing adaLN-Zero with a parameter-free rotation-based time conditioning.

| Models | #Layers | Hidden Size | MLP | #Heads | #Params |
|--------|---------|-------------|-------|--------|---------|
| GPDiT-B | 12 | 768 | 3072 | 12 | 85M |
| GPDiT-H | 24 | 2816 | 11264 | 22 | 2B |

We further scale the model to a two-billion-parameter variant, GPDiT-H (see Table 1). First, we perform a 200k-iteration warm-up using an unconditioned image dataset from LAION-Aesthetic [35] with a learning rate of 1e-4 and batch size of 960. Training continues for another 200k iterations on a mixed image-video dataset, with equal sampling of images and videos, and batch sizes of 256 and 64, respectively. Video frames are sampled every three frames and clipped into 17-frame segments. Each image is center-cropped to the resolution closest to the original, with target sizes of 256 × 256, 192×320, or 320×192, and video to 192×320. Finally, we continue training the GPDiT-H model on a pure video dataset featuring variable video lengths ranging from 17 to 45 frames. This stage lasts for an additional 150k iterations, using a reduced learning rate of 2e-5. The resulting model is denoted as GPDiT-H-LONG. To compress video latents, we employ WanVAE [43], which reduces four frames into a single latent representation.

## 5.2    Video Generation

To evaluate the generalization ability of the GPDiT framework, we conduct experiments on two zero-shot video generation tasks: MSRVTT and UCF-101 using GPDiT-H. The training data does not overlap with the test datasets, allowing us to assess the model's ability to generalize to unseen data. At the same time, to assess its fitting capability, we trained the GPDiT-B model on UCF-101

and measured its generation performance. For both models, 12-frame video sequences are generated, conditioned on 5 input frames. The generated results are evaluated using FID, FVD, and IS metrics. During inference, we apply classifier-free guidance with a scale of 1.2 for the GPDiT-H model and 2.0 for the GPDiT-B model.

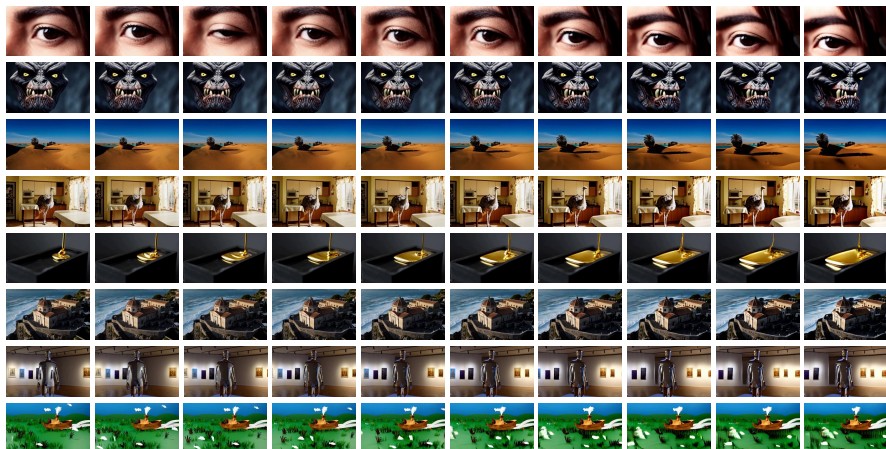

Figure 4: Video generation of the subsequent 16 frames conditioned on the initial 13 frames from the MovieGenBenchmark dataset, with the frames sampled at three-frame intervals thereafter. For more details, zoom in to observe finer aspects of the generation process.

Table 2: *Zero-Shot* performance comparison of video generation on MSRVTT and UCF-101, 12-frame video sequences are generated, conditioned on 5 input frames.

| Dataset | Method | #Data | #Params | FID ↓ | FVD ↓ | IS ↑ |
|---|---|---|---|---|---|---|
| MSRVTT | MagicVideo [55] | 10M | - | 36.5 | 998 | - |
| | LVDM [11] | 2M | 1.2B | - | 742 | - |
| | ModelScope [45] | 10M | 1.7B | - | 550 | - |
| | PixelDance [? ] | 10M | 1.5B | - | 381 | - |
| | DreamVideo [44] | 5.3M+340k | 2.0B | - | 149 | - |
| | SnapVideo [? ] | 1.1B | 3.9B | 8.5 | 110 | - |
| | GPDiT-H | 192M Img + 6.4M Vid | 2.0B | **7.4** | **68** | - |
| | GPDiT-H-LONG | 24M Vid | 2.0B | **7.4** | **64** | - |
| UCF-101 | MagicVideo [55] | 10M | - | 145.0 | 699 | - |
| | CogVideo [15] | 5.4M | 9B | - | 626 | 50.5 |
| | InternVid [47] | 28M | - | 60.3 | 617 | 21.0 |
| | Video-LDM [2] | 10M | 4.2B | - | 551 | 33.5 |
| | Make-A-Video [36] | 20M | 9.7B | - | 367 | 33.0 |
| | SnapVideo [? ] | 1.1B | 3.9B | 39.0 | 260 | - |
| | PixelDance [? ] | 10M | 1.5B | 49.4 | **243** | 42.1 |
| | GPDiT-H | 192M Img + 6.4M Vid | 2.0B | **14.8** | **243** | **66.5** |
| | GPDiT-H-LONG | 24M Vid | 2.0B | **7.9** | **218** | **66.6** |

**Main results.** Table 2 shows GPDiT achieves a competitive FID of 7.4 and an FVD of 68 on MSRVTT, demonstrating its effectiveness in handling diverse video generation tasks without direct exposure to the test data. Moreover, GPDiT consistently outperforms previous methods in both FID and FVD, underscoring its potential to handle a wide range of unseen video data. On UCF-101, GPDiT also performs well across metrics, with an IS of 66.5, FID of 14.8, and FVD of 243. Notably, GPDiT-H-LONG, trained with 24M video data, achieves the best results with an IS of 66.6, FID of 7.9, and FVD of 218, further showcasing the model's generalization ability. As shown in Table 3, both GPDiT-B-OF2 and GPDiT-B-OF achieve strong alignment with the UCF-101 distribution, attaining competitive FVD scores of 214 and 216 respectively with only 80M parameters. These results validate GPDiT's effectiveness in distribution fitting and its consistent robustness across varying model scales. For visual demonstration, we present the generated videos derived from 13 input frames alongside their corresponding 16-frame extensions on the MovieGenBench dataset [? ], as shown in Figure 4.

## 5.3 Video representation

To assess the model's representation ability, we conduct linear probing experiments with two attention mechanisms, extracting features from various layers of GPDiT-B and GPDiT-H. It is important to note that GPDiT-B is trained on UCF-101, while GPDiT-H is trained on close-source open-domain dataset, so the representation ability measured is both fit and generalized. The probing task is constructed by globally pooling features extracted from the frozen GPDiTmodel and training a logistic layer for the UCF-101 classification task. For each sample, we uniformly select 13 frames, spaced three frames apart, and pass them through the backbone without temporal rotation.

**Main results.** Figure 5a shows the classification accuracy for two different attention mechanisms on the GPDiT-B model. Notably, OF2 outperforms OF by a significant margin, highlighting the enhanced representation performance when allowing interactions between clean context frames. This aligns with intuition, as interactions between clean frames enhance the model's ability to understand the content. We also observe that the classification accuracy tends to peak at earlier layers, initially rising at shallow layers before gradually declining. This is consistent with the classification results presented in REPA [53], where enhanced representation ability strengthens fitting in the shallow layers. This further demonstrates GPDiT's ability to fit and

Table 3: FVD comparison of methods trained on UCF-101.

| Model | #Params | Type | FVD |
|---|---|---|---|
| ExtDM-K2 [54] | 119 M | Video-DIT | 394 |
| OmniTokenizer [? ] | 227M | Token-AR | 314 |
| ACDiT [? ] | 130M | Frame-AR | 376 |
| MAGI [? ] | 850M | Frame-AR | 297 |
| FAR [10] | 130M | Frame-AR | 194 |
| GPDiT-B-OF | 80M | Frame-AR | 216 |
| GPDiT-B-OF2 | 80M | Frame-AR | 214 |

improve representation quality. Figure 5b illustrates the classification accuracy of GPDiT-H-OF2 across different training steps and layers. As training progresses, the classification accuracy steadily improves. Additionally, since GPDiT-H-OF2 is zero-shot on the UCF-101 dataset, accuracy peaks at the 2/3 layer, which is inconsistent with the results of GPDiT-B. Figure 5c demonstrates the correlation between the generation metric (FVD) and classification accuracy for GPDiT-H-OF2. There is a clear positive correlation between generative capability and representational ability, indicating that as training progresses, both the generative performance and the representation ability of GPDiT improve simultaneously.

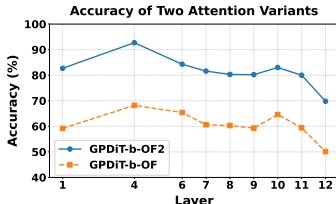 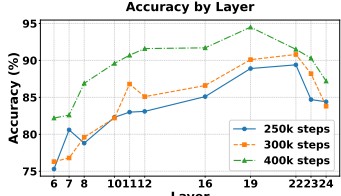 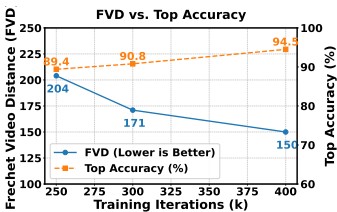

(a) Classification accuracy of GPDiT-B between OF and OF2 attention mechanisms across different layers.

(b) Classification accuracy of GPDiT-H-OF2 across different layers and training iterations.

(c) FVD of GPDiT-H-OF2 and Top accuracy across training steps.

Figure 5: Linear probing performance of GPDiT across different training settings.

## 5.4 Video Few-shot Learning

The pre-trained GPDiT exhibits strong representational ability, and our AR paradigm enables conditioning via sequence concatenation, allowing easy generalization to other tasks without the need for additional modules like VACE [18] or IP-Adapter [51]. This motivates the investigation of the few-shot learning capabilities of pre-trained models across multiple tasks, including grayscale conversion, depth estimation, human detection, image colorization, canny edge-to-image reconstruction, and two style transfer applications. The pretrained GPDiT-H model undergoes 500 iterations of fine-tuning with a batch size of 4, optimized to generate transformations conditioned on both input images and

contextual demonstrations. During testing, the model uses two (source, target) pairs as dynamic conditioning inputs to generate the transformed output for an unseen source image.

**Main results.** Figure 6 and Figure 7 show that GPDiT is able to transfer to multiple downstream tasks after few-shot learning. It is clearly demonstrated that GPDiT can easily convert color images to black-and-white and vice versa. In the Human Detection task, the model accurately distinguishes the number of people and their body skeleton. Additionally, it supports controllable editing by generating controlled instances using edge maps. For instance, Figure 7 shows that the bird generated in the Canny Edge to Image task follows the contours with fine detail. We also explored popular style transfers, such as TikTok-style face-to-cartoon transformations and GPT4o-Ghibli art style switches (Figure 7). Additionally, since only 20 shots are needed for few-shot learning, similar to GPT-2, this suggests the potential for larger GPDiT models to exhibit emergent In-Context Learning (ICL) abilities, as seen in the transition from GPT-2 to GPT-3.

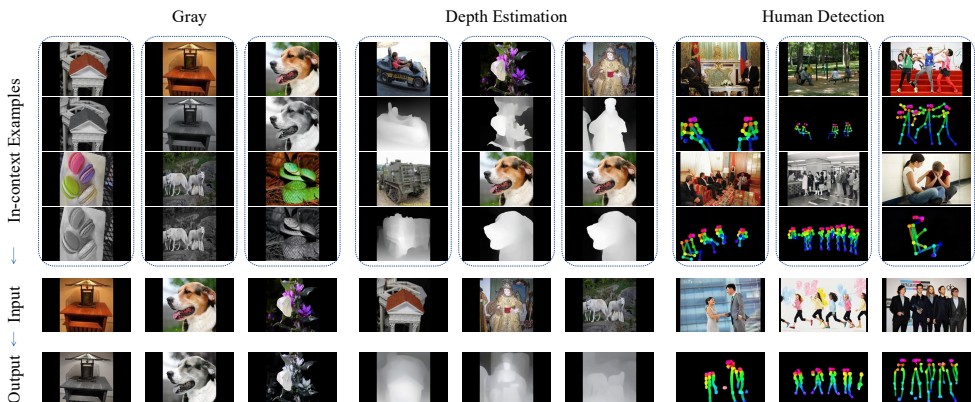

Figure 6: Extracting Low Information from Images: Skeleton, Depth, and Grayscale Transformations.

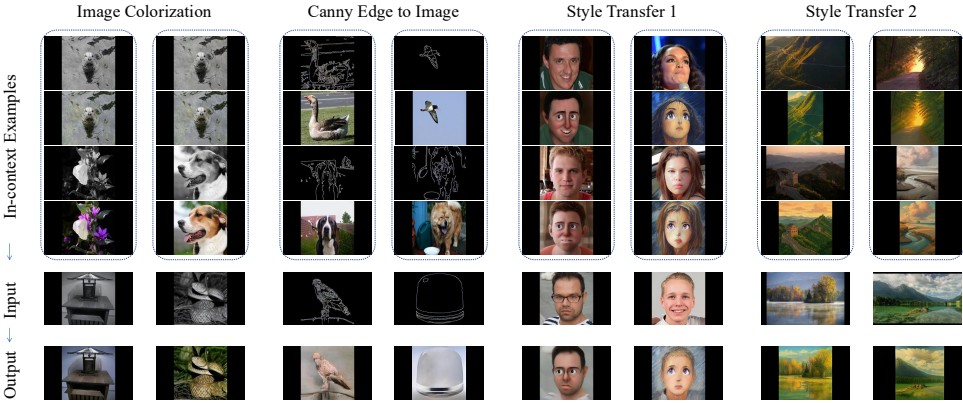

Figure 7: Results of Style Transfer and Conditional Generation: Image Colorization, Edge-to-Image and Two Style Transfer Tasks.

## 6 Discussion

In this work, we present a novel framework that unifies autoregressive modeling and diffusion for video generation. Our method incorporates a lightweight attention mechanism that leverages temporal redundancy to reduce computational overhead, as well as a parameter-free, rotation-based time-conditioning strategy to efficiently inject temporal information. These design choices enable faster training and inference without sacrificing model performance. Extensive experiments demonstrate that our model achieves state-of-the-art performance in video generation, competitive results in video representation, and robust generalization in few-shot multi-task settings, underscoring its versatility across various video modeling tasks.

**Limitations.** Due to resource constraints, we are unable to scale our experiments to larger configurations. In future work, we plan to explore larger-scale models and investigate their potential for in-context learning. While our model demonstrates strong representational capabilities, it is currently limited to the video modality. The design choice of conditioning along the sequence dimension enables seamless integration of other modalities, allowing natural extensibility to multi-modal inputs, such as language. We intend to explore this unified generative-understanding model in future research.

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

# A Additional Experimental Details and Results

## A.1 Video generation results

We begin by evaluating the effect of our lightweight attention variant on training convergence, in comparison to standard causal attention. As shown in Figure 8, the OF2 model exhibits a slight improvement in training convergence speed compared to the OF one.

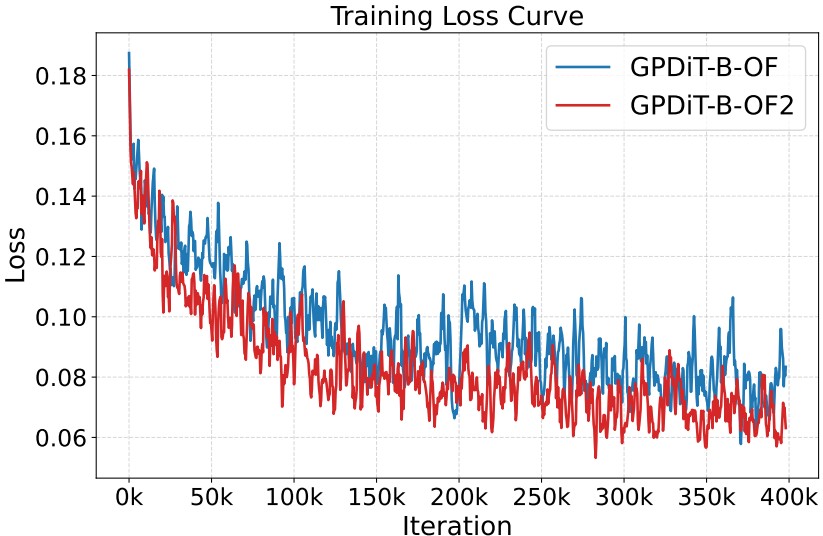

Figure 8: Training loss comparison of GPDiT-B-OF2 and GPDiT-B-OF.

Figure 9 and Figure 10 showcase video prediction results conditioned on camera motion. Each example is generated from 13 input frames and extended by 16 predicted frames, sampled at every fourth frame from the MovieGenBench dataset. The top row presents the ground truth, and the bottom row shows the predicted frames. These examples demonstrate the model's ability to accurately anticipate dynamic camera movements and physical scene transitions.

## A.2 Visual Question Answering results

To further validate the discriminative capability of our model's representations, we replace the vision encoder pretrained CLIP in Video-ChatGPT [29] with the first 18 layers of our trained GPDiT-H model as the feature extractor. Correspondingly, we replace the original linear layer with a new linear layer matching the output dimensions of GPDiT-H. Following the standard practice of Video-ChatGPT, we freeze the parameters of the feature extractor and train only the linear layer. The training batch size is set to 4, the learning rate is set to 2e-5, and the number of training epochs is 6.

It is worth noting that our model is not trained in language modality and solely based on video modality for representation learning. Despite this, as illustrated in Table 4, our model achieves an accuracy of 28.2% on ActivityNet-QA, compared to the original 35.2%. This result further validates the competitive discriminative capability of our model's representations, demonstrating promising potential for future multi-modality integration and training.

Table 4: Activity Net-QA

| Model | Accuracy | Score |
|---|---|---|
| Video-ChatGPT | 35.2 | 2.8 |
| GPDiT-H-OF2-Long | 28.2 | 1.54 |

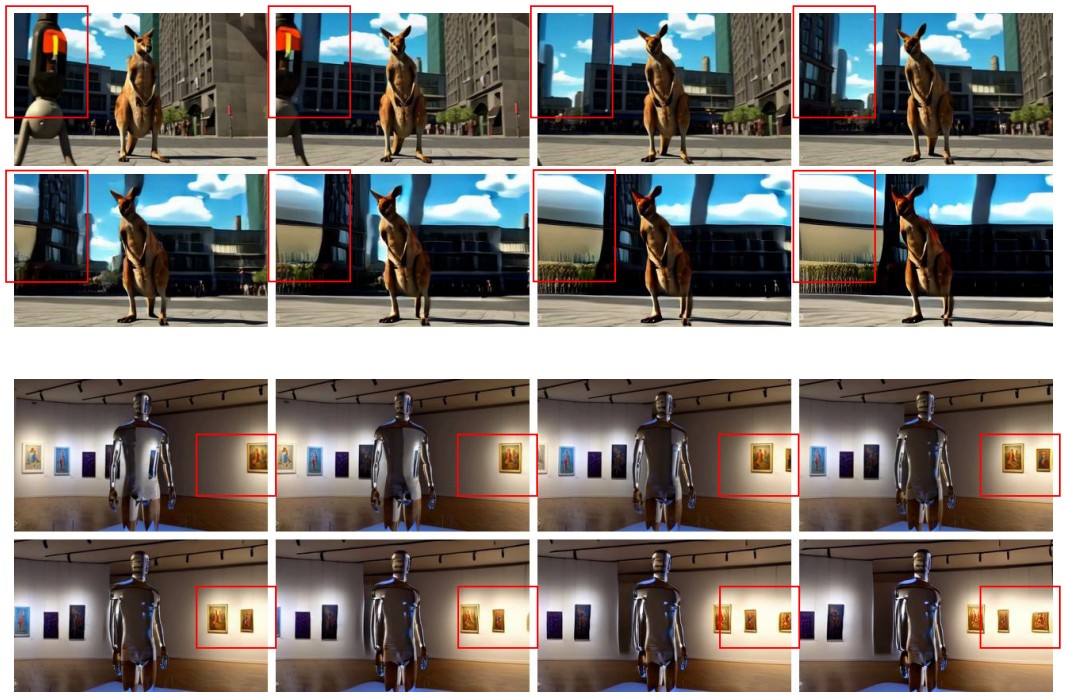

Figure 9: Camera Motion Control Prediction (Camera Rotation).

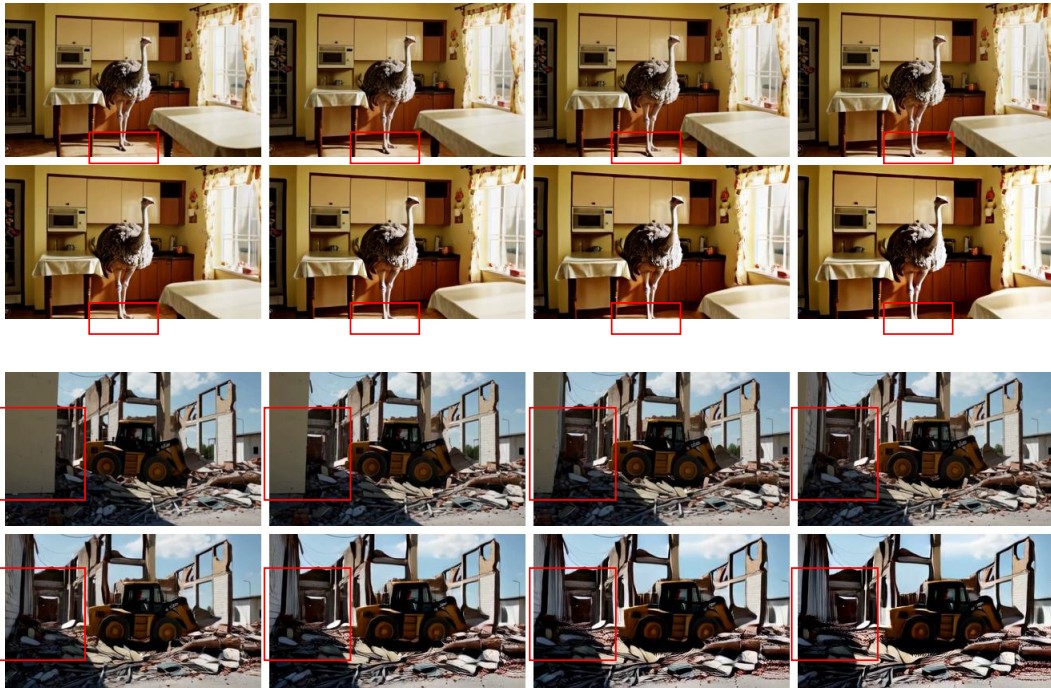

Figure 10: Camera Motion Control Prediction (First: Far-to-Near, Second: Left-to-Right).

## A.3 More Video Few-shot Learning results

We conducted two sets of experiments to evaluate the model's generalization ability. In the first setting, generalization from a known style to the real-face domain, the model was trained on 11 face-style-to-real-face translation tasks using approximately 20 video samples per style. During testing, images from one of the 11 styles seen during training were introduced, with no overlap between

the training and test sets. After around 20 epochs of training, the model successfully generated realistic face outputs conditioned on previously unseen test images, demonstrating strong intra-style generalization. In the second setting, few-shot generalization from real faces to novel style, the model was trained on real-face-to-10-style mappings and evaluated on one previously unseen target style. During inference, we provided the corresponding style conditions for this novel style, and the model was able to synthesize outputs that accurately reflected the unseen target style, highlighting its few-shot generalization capability and potential for in-context learning.

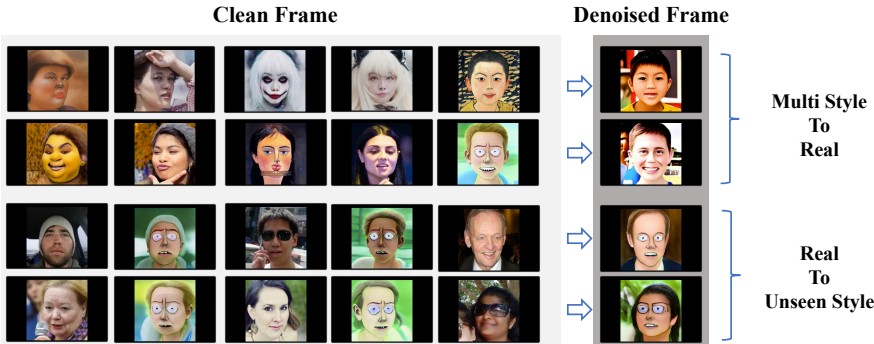

Figure 11: Qualitative results demonstrating generalization from multiple styles to the real-face domain and few-shot adaptation from real faces to unseen style.

