# OpenReview forum: "Generative Pre-trained Autoregressive Diffusion Transformer"
_NeurIPS.cc/2025/Conference — NeurIPS 2025 poster_

### Official Review · Reviewer_FKsG · 2025-06-24

**Clarity:** 3
**Significance:** 2
**Originality:** 3
**Rating:** 5
**Confidence:** 4

**Summary:**

This paper introduces GPDiT to unify the advantages of diffusion and autoregressive modeling for long-range video generation task. GPDiT autoregressively produces next latent frames through diffusion loss, while achieving plausible modeling of motion dynamics and consistency across frames. The proposed method improves both generation quality and representation capabilities. Moreover, it also presents a lightweight causal attention and parameter-free rotation-based time-conditioning mechanism to improve training and inference efficiency. Comprehensive experiments show that GPDiT obtains superior performance in video generation, video representation, and few-shot learning tasks.

**Questions:**

1. For Video Few-shot Learning, how many training samples are required for these tasks to acquire in-context learning capabilities, and are there differences between tasks of varying difficulty levels?
2. Has the author attempted more complex in-context learning tasks, such as image editing tasks?
3. The method in this paper is implemented based on Wan's VAE, which compresses multi-frame video frames into a single latent. Then, when migrating to image tasks, how are these images encoded?

**Ethical Concerns:**

["NO or VERY MINOR ethics concerns only"]

**Final Justification:**

My concerns have been well addressed by the rebuttal, so I am glad to increase my rating to 5.

**Limitations:**

Yes

**Quality:**

3

**Strengths And Weaknesses:**

**Strengths**
1. This paper introduces GPDiT with framewise causal attention to improve temporal consistency of long videos, as well as a lightweight causal attention to reduce computational costs.
2. This paper presents a rotation-based conditioning strategy to provide a parameter-free approache to inject time information.
3. Extensive experimental results show that the proposed GPDiT obtains comparable performance on video generation and effective extension in few-shot learning tasks.

**Weaknesses**
1. Lack of comparison with the baselines for popular video generation. The authors only provided comparisons with earlier methods, and are encouraged to include more advanced methods, such as CogVideo-X, HunyuanVideo, Wan.
2. The core contribution of this paper is the proposal of "Lightweight Causal Attention", but the authors did not quantitatively explore the training and sampling efficiency of this technique.
3. There is a lack of exploration on "injecting time information". The authors proposed a new "Timestep Conditioning Injection" method in Sec 4.2, but did not conduct ablation studies on this technique.

---

> ### Author Rebuttal · Authors · 2025-07-31
>
> Thank you for taking the time to review our manuscript. We greatly appreciate your valuable feedback. Below are our point-by-point responses to your comments:
>
> > ### W1: Comparison with CogVideo-X, Hunyuan-Video, Wan.
>
> As our paper primarily focuses on video-to-video gen-
> eration, we do not include text-to-video (T2V) meth-
> ods such as HunyuanVideo and Wan as baselines.
> However, we further extend our method to the T2V
> setting, and the results are presented in W1 to Reviewer g1aP.
>
> As our method integrates autoregressive and diffusion components, we conduct a thorough comparison with recent AR-diffusion hybrid baselines, including FAR[1], MAGI[2] and ACDIT[3]. These baselines represent the most recent advances in the field at the time of our study. As NOVA[4] supports video-to-video generation, we will include it as a baseline in the revised version to provide a more comprehensive comparison.
>
> [1]Long-context autoregressive video modeling with next-frame prediction. arXiv preprint arXiv:2503.19325, 2025.
>
> [2]Taming teacher forcing for masked autoregressive video generation. Proceedings of the Computer Vision and Pattern Recognition Conference. 2025.
>
> [3]Acdit: Interpolating autoregressive conditional modeling and
> diffusion transformer. arXiv preprint arXiv:2412.07720, 2024.
>
> [4]Autoregressive video generation without vector
> quantization. arXiv preprint arXiv:2412.14169, 2024.
>
>
>
> > ### W2: Quantification of lightweight attention's practical speedup
>
>
> We conducted experiments on the GPDiT-B model to evaluate the inference efficiency of our proposed method. To evaluate inference efficiency, we configured the denoising process with 30 steps and generated videos with 33 frames. The following table reports the average time required to generate a single video under this setting.
>
> #### Table 1: empirical validation of effectiveness on GPDiT-B
> | Method              | inference time (second/iter)|
> | ------------------ | --------------------- |
> | O(F) |     3.34             |
> | O(F^2)| 3.68   |
>
>
> > ### W3: Lack of ablation study
>
> To evaluate the effectiveness of our proposed rotation-based time injection strategy, we conduct an ablation study comparing it with standard AdaLN-Zero and AdaLN-Single [1]. The following results demonstrate that the rotation-based strategy achieves better memory efficiency while maintaining comparable performance to the baseline methods.
>
>
> #### Table 2: Ablation of time-conditioning mechanism  with GPDiT-B on ConceptualCaption12m and Panda7m Dataset
> | Method   | Metric | 10K  | 15K  | 20K  | 25K  |
> |----------|--------|------|------|------|------|
> | Rotation | Loss      | 0.194 | 0.184 | 0.184 | 0.183 |
> |          | Memory | 34G | 34G | 34G | 34G |
> |               | MaxGradNorm | 0.313 | 0.281 | 0.259 | 0.259 |
> | adaLN-Zero    | Loss   | 0.178 | 0.173 | 0.176 | 0.175 |
> |               | Memory | 36G | 36G | 36G | 36G |
> |               | MaxGradNorm | 0.229 | 0.219 | 0.216 | 0.217 |
> | adaLN-single  | Loss   | 0.185 | 0.182 | 0.177 | 0.174 |
> |               | Memory | 34.6G | 34.6G | 34.6G | 3.3G |
> |               | MaxGradNorm | 0.291 | 0.281 | 0.276 | 0.267 |
>
>
>
> [1]Pixart-α: Fast training of diffusion transformer for
> photorealistic text-to-image synthesis. arXiv preprint arXiv:2310.00426, 2023.
>
>
> > ### Q1: Video Few-shot Learning
>
> For video few-shot learning, we sample 20 examples as the training set and additional non-overlapping examples as the test set. Each task is trained for only 500 iterations. This setting ensures a fair evaluation of the model’s generalization ability on unseen data. For tasks such as grayscale conversion, depth estimation, and human detection, the models converge quickly within 100 iterations. In contrast, the two style transfer tasks converge more slowly compared to the above tasks.
>
> > ### Q2: More complex in-context learning tasks
>
> While our current work does not cover complex in-context learning tasks, we believe that our model can be extended to more challenging scenarios by incorporating more diverse training data. We plan to evaluate more complex in-context learning tasks and include representative examples in our revised manuscript.
>
> > ### Q3: How to encode image
>
> During training, we add the padding to the image to adjust the comreession ratio of Wan's VAE. In our few-shot learning setting, we replicate the single input image four times to construct a 5-frame sequence before feeding it into the VAE.

---

### Official Review · Reviewer_kYES · 2025-06-28

**Clarity:** 1
**Significance:** 2
**Originality:** 2
**Rating:** 4
**Confidence:** 4

**Summary:**

This paper introduces GPDiT, A unified framework for video generation that combines autoregressive next-token prediction objective with a diffusion-based formulation, diverging from the more common discrete token-based approaches. GPDiT enhances model efficiency during both training and inference through a specialized causal attention mechanism that strategically bypasses attention score computation for (clean) input frames. Furthermore, The authors also propose a parameter-free rotational time conditioning mechanism as an alternative to adaLN-Zero layers used in prior DiT-based models. Two model variants are trained: GPDiT-B (12 layers, 85M parameters) and GPDiT-H (24 layers, 2B parameters) on UCF-101 and MSR-VTT, using sequences of 17 frames.Additionally, a longer variant, GPDiT-H-LONG, is trained with variable-length sequences (17–45 frames). Experimental results consistently demonstrate that GPDiT performs competitively or better than tested methods on both qualitative and quantitative metrics. Beyond video generation, GPDiT exhibits strong capabilities in learning robust representations, enabling various vision-based in-context learning tasks such as colorization, depth prediction, human pose detection, face-to-cartoon synthesis, and style transfer.

**Questions:**

1. Given that NOVA [7] can also perform image-to-video generation, the experimental evaluation section would greatly benefit from a direct quantitative and qualitative comparison against NOVA. What are the advantages and disadvantages of GPDiT compared to NOVA, especially considering the architectural differences.

2. Table 2 presents evaluations for 12-frame video generation conditioned on 5 input frames. What are the performance and visual quality implications if the models are conditioned on only a single input frame? Does the model maintain coherence and quality under this more challenging setting?

3. As an autoregressive model, the visual quality of generated frames might degrade over time. Could the authors provide an analysis or qualitative examples demonstrating how the visual quality of frames changes as the generation sequence extends, especially for longer video generations?

4. The proposed parameter-free rotation-based time conditioning is a core contribution. How does its effectiveness compare against the adaLN-single layer proposed in PIXART-α [6], particularly concerning performance, stability, and computational overhead?

5. The in-context learning experiments utilize models trained with custom-built SFT data featuring task-specific image pair samples. If the models are primarily trained on single-modality inputs (e.g., sequences of color images, grayscale images, or depth images), how well do they generalize and perform few-shot in-context learning when presented with mixed input types within the same context? This would explore the model's ability to learn and apply representations across different visual modalities.

**Ethical Concerns:**

["Major Concern: Data privacy, copyright, and consent"]

**Final Justification:**

There are still some small issues regarding the evaluation of some components, but the authors provided extra experiment and comment to most of my questions. Hence, I am now leaning towards acceptance.

**Limitations:**

The authors have adequately discussed the limitations of their approach.

**Paper Formatting Concerns:**

I did not observe any major formatting issues in the paper.

**Quality:**

2

**Strengths And Weaknesses:**

**Strengths**
- A significant strength of this work is the extensive evaluation of the proposed models beyond just video generation. The authors demonstrate the efficacy of GPDiT in learning high-quality video representations (validated through classification tasks) and its remarkable in-context learning abilities across a diverse range of vision tasks. This highlights the framework's versatility and potential for broader applications.
- The introduction of a parameter-free, rotation-based time conditioning mechanism is a particularly interesting contribution. Its practical effectiveness, as evidenced by the experimental results, suggests a promising direction for more efficient and robust temporal conditioning in diffusion models.

**Weaknesses**
- While the underlying formulation of GPDiT is inherently extensible to text-to-video generation, the current work is solely trained and evaluated on visual data. Consequently, the authors are restricted to quantitative metrics like FVD, FID, and IS. This omission prevents a direct comparison with state-of-the-art text-to-video models and misses out on recently proposed and widely adopted prompt-based benchmarks such as VBench and MovieGenBench. Addressing text-to-video capabilities would significantly broaden the paper's impact.
- The related work section lacks a comprehensive discussion distinguishing GPDiT from highly relevant text-to-video models such as NOVA [7] and CausVid. These models also leverage diffusion losses within an autoregressive framework. A more detailed comparison highlighting the architectural novelties, efficiency gains, or specific contributions of GPDiT over these similar approaches is essential for contextualizing the paper's advancements and demonstrating its unique value.

CausVid:  Yin et al., From Slow Bidirectional to Fast Autoregressive Video Diffusion Models, CVPR 2025

- The paper contains several vague and inconsistent experimental details that hinder clarity and reproducibility:

(i) Line 236 states that "GPDiT-H is trained on close-source open-domain dataset," which contradicts the earlier impression that models are trained exclusively on UCF-101 and MSR-VTT. Clarification on the specific datasets used for each model version is needed.

(ii) There is an unexplained discrepancy between the qualitative results for video generation (16 frames conditioned on 13 initial frames) and the quantitative results (12-frame generations conditioned on 5 input frames). A clear rationale for these differing evaluation setups should be provided.

**Minor Points**
- Figure 3 caption: “OF^2 => O(F$^2$) and OF => O(F)

---

> ### Author Rebuttal · Authors · 2025-07-31
>
> Thank you for taking the time to review our manuscript. We greatly appreciate your valuable feedback. Below are our point-by-point responses to your comments:
> > ### W1: Text-to-Video
>
> Due to NeurIPS submission restrictions, we are unable to include visualizations of our results. Instead, we provide the loss vs. epoch metrics for the text-to-video (T2V) experiments. Specifically, we report the training loss across epochs for GPDiT-H (2B parameters).
> The steadily decreasing loss indicates the potential of our model to generalize effectively to T2V tasks. The detailed results are presented in the table below.
>
> #### Table 1: Text-to-Video Generation with GPDiT-H on the ConceptualCaption12m and Panda7m Dataset
>
>
> | Iteration | 500 | 1000 | 1500 | 2000 | 2500 | 3000|
> |-----------|-----|-----|-----|-----|-----|-----|
> | Loss      | 0.168 | 0.142 | 0.142 | 0.138 |0.136| 0.139|
>
>
>
> > ### W2: Comprehensive discussion with other works
>
> As our method integrates both autoregressive and diffusion components, we conduct a thorough comparison with recent AR-diffusion hybrid baselines, including FAR [1], MAGI [2], and ACDIT [3], which represent the latest advances in this area at the time of our study.
>
> While we acknowledge that a discussion of similarities and differences with NOVA[4] and CausalVid[5] would further highlight the advancements and unique contributions of our method, we note that both NOVA and CausalVid mainly target the text-to-video task. In particular, CausalVid relies on a separate bidirectional teacher model and employs asymmetric distillation to train a causal student. In contrast, our method is entirely self-contained and trained from scratch, requiring no teacher supervision. As NOVA supports video-to-video generation, we will include it as a baseline in the revised version to provide a more comprehensive comparison.
>
> [1]Long-context autoregressive video modeling with next-frame prediction. arXiv preprint arXiv:2503.19325, 2025.
>
> [2]Taming teacher forcing for masked autoregressive video generation. Proceedings of the Computer Vision and Pattern Recognition Conference. 2025.
>
> [3]Acdit: Interpolating autoregressive conditional modeling and
> diffusion transformer. arXiv preprint arXiv:2412.07720, 2024.
>
> [4]Autoregressive video generation without vector
> quantization. arXiv preprint arXiv:2412.14169, 2024.
>
> [5]CausVid: Yin et al., From Slow Bidirectional to Fast Autoregressive Video Diffusion Models, CVPR 2025
>
>
> > ### W3: Clarity and reproducibility
>
> As stated in lines 213–214, GPDiT-H is trained on a closed-source open-domain dataset and evaluated in a zero-shot setting on UCF-101 and MSR-VTT, with no overlap between training and test data. Our closed-source dataset is constructed by mixing several publicly available datasets, including Panda-70M, Mixkit, Pixabay, and Pexels for video data, and LAION-400M for image data. We further apply a series of data filtering and cleaning procedures to ensure data quality and relevance.
> Regarding experimental alignment, since our WanVAE[1] compresses 4n+1 frames into n+1 latent frames, our c5p12 setting aligns with c4p12 in FAR[2]. The c13p16 setting is used to explore performance under longer-frame scenarios.
> We will open-source our code and models, along with detailed instructions for reproduction.
>
> [1]Wan: Open and advanced large-scale video generative
> models. arXiv preprint arXiv:2503.20314, 2025.
>
> [2]Long-context autoregressive video modeling with next-frame prediction. arXiv preprint arXiv:2503.19325, 2025.
>
>
> > ### Q1: Text-to-Video
>
> see Weakness 1
>
> > ### Q2: Visual Quality under single frame
>
>  In this case, we observe that the generated video tends to remain static for the initial frames. Starting from around the 17th frame, the video typically begins to degrade, leading to incoherent or implausible results.
>
> > ### Q3: Long video generation
>
> We also evaluated our model on longer video generation tasks (up to 70 frames). We observed that while the model is capable of generating coherent content during the early stages, noticeable degradation in visual quality begins to appear around the 40th frame. This degradation becomes increasingly severe, and by the 50th frame, the generation typically collapses, resulting in implausible or incoherent outputs. Nevertheless, considering that our model was only trained on sequences of up to 33 frames, these results demonstrate a certain degree of extrapolation capability beyond the training horizon.
>
> > ### Q4: Ablation study on time injection strategy
> To evaluate the effectiveness of our proposed rotation-based time injection strategy, we conduct an ablation study comparing it with standard AdaLN-Zero and AdaLN-Single [1]. The following results demonstrate that the rotation-based strategy achieves better memory efficiency while maintaining comparable performance to the baseline methods.
>
>
> #### Table 2: Ablation of time-conditioning mechanism  with GPDiT-B on ConceptualCaption12m and Panda7m Dataset
> | Method   | Metric | 10K  | 15K  | 20K  | 25K  |
> |----------|--------|------|------|------|------|
> | Rotation | Loss      | 0.194 | 0.184 | 0.184 | 0.183 |
> |          | Memory | 34G | 34G | 34G | 34G |
> |               | MaxGradNorm | 0.313 | 0.281 | 0.259 | 0.259 |
> | adaLN-Zero    | Loss   | 0.178 | 0.173 | 0.176 | 0.175 |
> |               | Memory | 36G | 36G | 36G | 36G |
> |               | MaxGradNorm | 0.229 | 0.219 | 0.216 | 0.217 |
> | adaLN-single  | Loss   | 0.185 | 0.182 | 0.177 | 0.174 |
> |               | Memory | 34.6G | 34.6G | 34.6G | 3.3G |
> |               | MaxGradNorm | 0.291 | 0.281 | 0.276 | 0.267 |
>
>
> [1]Pixart-α: Fast training of diffusion transformer for
> photorealistic text-to-image synthesis. arXiv preprint arXiv:2310.00426, 2023.
>
>
> > ### Q5: Mixed input types for few-shot learning
>
> Thank you for your question. We are not entirely sure we understand your point — could you please clarify it in more detail? Regarding few-shot learning, if a task is not included during training, the model generally lacks the ability to perform it effectively at test time.

---

> > ### Comment · Reviewer_kYES · 2025-08-02
> >
> > I appreciate the authors’ thoughtful and thorough responses to the concerns I raised. Several of my reservations have been satisfactorily addressed. However, I would like to follow up on a few points for further clarification:
> >
> > **(1) Impact of Input Context Length on Performance:** Could the authors elaborate on how varying the input context length affects overall model performance? Specifically, it would be helpful to see how the FVD (Fréchet Video Distance) metric changes as the input size varies from 1 to 5, or more.
> >
> > **(2) Comparison of Time-Conditioning Mechanisms:** Regarding the proposed rotation-based time-conditioning mechanism, could the authors provide quantitative results comparing its performance to traditional learned conditioning methods? The current table only shows the loss, memory and maxgradnorm, but not the FID, FVD, IS scores.
> >
> > **(3) Few-Shot Learning and Generalization:** Regarding my earlier comment on few-shot learning experiments, my main curiosity lies in evaluating the model’s ability to generalize to previously unseen modalities during the supervised fine-tuning (SFT) stage. Could the authors clarify whether the model can produce meaningful predictions for a novel modality without being explicitly fine-tuned on that modality at test time?

---

> ### Author Response · Authors · 2025-08-07
>
> We sincerely thank you for taking the time to review our response and your follow-up question. Below are our point-by-point responses to your comments:
>
> > ###  Q1 Impact of Input Context Length on Performance:
> To assess how input context length impacts overall model performance, we conducted additional experiments varying the number of input (conditioning) frames from 1 to 17.
> | Input Frames | Output Frames | FVD ↓       |
> |--------------|----------------|-------------|
> | 1            | 17             | 144.9961     |
> | 5            | 21             | 103.8084     |
> | 9            | 25             | 103.6789     |
> | 13           | 29             | 117.5831     |
> | 17           | 33             | 127.7384     |
>
> These findings support the intuition that moderate temporal context can significantly enhance generation quality, but further increasing it does not necessarily yield proportional benefits and may even slightly degrade performance.
>
> > ###  Q2 Comparison of Time-Conditioning Mechanisms:
>
> We feel regretful that, owing to limited computational resources, the current models have not yet fully converged. Consequently, reporting FVD, FID, or IS scores at this point would not be meaningful.
>
>
> > ### Q3 Few-Shot Learning and Generalization:
>
> We thank the reviewer for the insightful follow-up regarding few-shot learning and the model’s ability to generalize to previously unseen modalities during the supervised fine-tuning (SFT) stage. To assess this, we conducted two sets of experiments:
>
> 1. Generalization from a novel style to the real-face domain:
> We trained the model on 11 face-style-to-real-face translation tasks, with approximately 20 video samples per style. At test time, we introduced a 12th, previously unseen face style and provided its corresponding style condition. The model, trained for around 20 epochs, was able to generate plausible real-face outputs conditioned on the novel style.The results confirm that the model can generalize to unseen style inputs without explicit fine-tuning.
>
> 2. Few-shot generalization from real faces to novel styles:
> We trained the model on real-face-to-10-style mappings and tested on two previously unseen target styles. During inference, we supplied the corresponding style conditions for the novel styles. The model successfully generated outputs that reflected the unseen target styles, demonstrating its few-shot generalization capability and suggesting its potential for in-context learning.
>
> These results highlight the model’s ability to generalize beyond the training modalities, handling both unseen input styles and unseen output styles through conditioning alone. Detailed visualizations and quantitative results will be included in the final version of this work.

---

> > ### Comment · Reviewer_kYES · 2025-08-08
> >
> > I appreciate the authors’ efforts during the discussion phase. While some of my concerns have been partially addressed, a few points still require clarification:
> >
> > 1. Could you elaborate on the results related to input context length by providing some additional context or interpretation?
> > 2. For completeness, could you also include the quantitative quality scores across iterations for the different time-conditioning mechanisms—similar to how you presented the loss, memory usage, and maxgradnorm?

---

> > > ### Author Response · Authors · 2025-08-09
> > >
> > > We sincerely thank you for taking the time to review our response and your follow-up question. Below are our point-by-point responses to your comments:
> > >
> > > > ### Q1 Experimental settings
> > >
> > > We trained the 2B model on clips mostly 17 frames (max 49). At inference we fix CFG = 1.0. Because the video VAE uses 4k+1 temporal compression, we downsample by 4 (treat every 4 decoded frames as one) before computing FVD. Each evaluation generates 16 frames. Since the training distribution is short (≈17 total frames), very long conditioning contexts exceed the model’s effective horizon, and we observe a performance drop at the longest conditioning lengths.
> > >
> > > > ### Q2 More metrics
> > >
> > > As noted earlier, the models are far from convergence; stable performance typically appears around **400k steps**. We are currently at **30k**, and due to compute and time limits we can evaluate only **5k** images per checkpoint at limited time, whereas the standard FID protocol uses **50k**. Consequently, the interim **FID** values are noisy and not representative.
> > >
> > > | Iter (k) | FID ↓ (AdaLN)         | FID ↓ (Rotation) | FID ↓ (AdaLN Single) |
> > > | -------------: | ---------------------------------: | ---------------------: | ------------------------: |
> > > |       10 |         48.14 |      56.01 |                52.84 |
> > > |       20 |         54.72 |      55.82 |                48.83 |
> > > |       30 |         44.20 |      57.34 |                49.92 |

---

### Official Review · Reviewer_xKDW · 2025-07-02

**Clarity:** 3
**Significance:** 3
**Originality:** 3
**Rating:** 4
**Confidence:** 3

**Summary:**

This paper introduces GPDiT, a novel hybrid framework that unifies autoregressive modeling and diffusion processes in a continuous latent space for long-range video synthesis. Its core contributions are:

1. Continuous Autoregressive Diffusion: Instead of discrete tokens, GPDiT predicts future latent frames autoregressively using a diffusion loss, enabling joint optimization of motion dynamics and semantic consistency across frames.
2. Efficiency Innovations:
   - A lightweight causal attention variant reduces computational overhead.
   - A parameter-free rotation-based time-conditioning mechanism enhances temporal modeling.
3. Multifaceted Capabilities: Experiments demonstrate GPDiT excels in:
   - Video generation quality (long-range consistency),
   - Representation learning (e.g., feature transferability),
   - Few-shot adaptation tasks.
The work establishes continuous latent space as a viable paradigm for scalable video modeling.

**Questions:**

1. Quantification of Efficiency Gains
   - Question: Lightweight attention claims "improved training/inference efficiency" but lacks metrics.
2. Statistical Reliability of Few-Shot Experiments
   - Question: Using merely 20 videos for few-shot evaluation (detection/colorization) risks overfitting. Was performance verified across data scales?
3. Generalizability of Rotation Conditioning
- Question: How does rotation injection handle non-square features (e.g., 768×320 videos)? Does complex-plane rotation require adaptation?

**Ethical Concerns:**

["NO or VERY MINOR ethics concerns only"]

**Limitations:**

YES.

**Paper Formatting Concerns:**

NO.

**Quality:**

3

**Strengths And Weaknesses:**

Strengths:

Quality: Comprehensive experimental design covering video generation (UCF-101/MSR-VTT), representation learning (linear probing accuracy), and few-shot multi-task applications (object detection/colorization/style transfer). Appropriate metrics (FVD/FID/IS) and ablation studies validate the efficacy of the lightweight attention and rotation-based conditioning.

Originality : Two innovative contributions:

1. Continuous latent space AR-diffusion framework: First unification of AR's causal modeling with diffusion's continuous optimization, resolving long-video semantic coherence (vs. discrete token AR models).
2. Zero-parameter time conditioning: Rotation injection maps noise to complex-plane rotation, replacing adaLN-Zero and significantly reducing parameters.

Significance: Proposes a unified video generation-understanding framework. Few-shot experiments demonstrate cross-task generalization potential, with multimodal extensibility (Discussion section).


Weaknesses:

- Quality Issues: Missing critical baselines: No comparison with state-of-the-art DiT variants or AR-diffusion hybrids, undermining SOTA claims.
- Significance Limitations: No quantification of lightweight attention's practical speedup, weakening efficiency claims.
- Clarity: adding algorithm pseudocode would further improve reproducibility.

---

> ### Author Rebuttal · Authors · 2025-07-31
>
> Thank you for taking the time to review our manuscript. We greatly appreciate your valuable feedback. Below are our point-by-point responses to your comments:
>
> > ### W1：Missing critical baselines
>
> As our method integrates autoregressive and diffusion components, we conduct a thorough comparison with recent AR-diffusion hybrid baselines, including FAR[1], MAGI[2] and ACDIT[3]. These baselines represent the most recent advances in the field at the time of our study. We emphasize that our evaluations are fair and do not undermine existing state-of-the-art claims.
> As NOVA[4] supports video-to-video generation, we will include it as a baseline in the revised version to provide a more comprehensive comparison.
>
> [1]Long-context autoregressive video modeling with next-frame prediction. arXiv preprint arXiv:2503.19325, 2025.
>
> [2]Taming teacher forcing for masked autoregressive video generation. Proceedings of the Computer Vision and Pattern Recognition Conference. 2025.
>
> [3]Acdit: Interpolating autoregressive conditional modeling and
> diffusion transformer. arXiv preprint arXiv:2412.07720, 2024.
>
> [4]Autoregressive video generation without vector
> quantization. arXiv preprint arXiv:2412.14169, 2024.
>
> > ### W2 & Q1: Quantification of lightweight attention's practical speedup
>
> We conducted experiments on the GPDiT-B model to evaluate the inference efficiency of our proposed method. To evaluate inference efficiency, we configured the denoising process with 30 steps and generated videos with 33 frames. The following table reports the average time required to generate a single video under this setting.
>
> #### Table 1: empirical validation of effectiveness on GPDiT-B
> | Method              | inference time (second/iter)|
> | ------------------ | --------------------- |
> | O(F) |     3.34             |
> | O(F^2)| 3.68   |
>
>
> > ### W3: Algorithm pseudocode
>
> To improve clarity and reproducibility, we will include the algorithm pseudocode in the revised version.
>
> > ### Q2: Statistical Reliability of Few-Shot Experiments
>
> We ensure that there is no overlap between the training and testing sets. In few-shot scenarios, using fewer examples aligns with the intended setup, while a larger dataset overall tends to improve model generalization and robustness.
>
> > ### Q3: Generalizability of Rotation Conditioning
>
> For rotation conditioning, we apply the rotation along the feature dimension (i.e., the last dimension). As a result, it does not affect features with non-square shapes.

---

> > ### Author Response · Authors · 2025-08-09
> >
> > Dear Reviewers,
> >
> > Thank you for the thoughtful discussion and your time. As the rebuttal is closing, we would like to confirm whether our responses have addressed your concerns. If anything remains unclear or you have additional questions, please let us know. Your feedback is invaluable and will help us improve the work.
> >
> >
> > Best regards,
> >
> > The authors of Submission 15795

---

### Official Review · Reviewer_g1aP · 2025-07-03

**Clarity:** 3
**Significance:** 2
**Originality:** 3
**Rating:** 4
**Confidence:** 4

**Summary:**

This paper proposed a novel framework GPDiT that models AR diffusion for long-range video synthesis within a continuous latent space. GPDiT autoregressively predicts future latent frames using a diffusion loss. The paper highlights two key innovations: a lightweight causal attention variant that reduces computational costs by eliminating attention computation between clean frames during training, and a parameter-free rotation-based time-conditioning mechanism that reinterprets noise injection as a rotation in a complex plane, thus removing the need for adaLN-Zero and its associated parameters. Experiments demonstrate GPDiT's strong performance in video generation quality, video representation ability, and few-shot learning tasks, positioning it as an effective framework for continuous video modeling.

**Questions:**

See weaknesses.

**Ethical Concerns:**

["NO or VERY MINOR ethics concerns only"]

**Final Justification:**

The rebuttal addressed my concerns well. I keep my rating.

**Limitations:**

The authors acknowledge resource constraints limiting larger-scale experiments and the current model's limitation to the video modality, with plans for future multi-modal extensions. The streaming video generation for longer length is still hindered by error accumulation. The computation efficiency problem addressed by this paper is not that important.

**Quality:**

3

**Strengths And Weaknesses:**

Strengths
- The lightweight causal attention and parameter-free rotation-based time-conditioning is novel, addressing critical challenges in streaming video generation.
- The paper provides clear explanations and illustrations (Figure 2 and 3) of the proposed attention mechanisms and the rotation-based time conditioning. And provides a clear analysis of computational savings from the lightweight causal attention.
- The experimental results demonstrate competitive performance across multiple tasks: video generation, video representation (validated through linear probing), and few-shot learning (transferring to various downstream tasks with minimal fine-tuning), which presents GPDiT's potential as a unified model for visual understanding and generation.

Weaknesses
- While the paper mentions the potential for extensibility to multi-modal inputs like language, the current experiments are limited to video. The scale of the experiments (2B model on MSRVTT ) is not large enough. We cannot understand its potential in scaling up further. Showing scaling curves between the number of paramsters/tokens and accuracy is better.
- The paper lacks a blation study for the parameter-free rotation-based time-conditioning mechanism. A comparison against traditional learned conditioning methods (e.g., adaLN-Zero) essential to validate its effectiveness and efficiency claims.
- Although the paper discusses the theoretical computational efficiency improvements of the lightweight causal attention, it does not provide empirical validation (e.g., training time, inference speed, or memory usage comparisons) to support these claims.
- The core challenge in streaming video generation is the error accumulation. There are many other choices to improve the computation efficiency, e.g., sparse attention, consistency distillation, etc. This paper tackles a problem that is not that important. And whether the proposed strategy can be incorporated to the other acceleration strategies remains unknown.

---

> ### Author Rebuttal · Authors · 2025-07-31
>
> Thank you for taking the time to review our manuscript. We greatly appreciate your valuable feedback. Below are our point-by-point responses to your comments:
>
>
> > ### W1: Extension to Text-to-Video experiments and Scaling curves
>
> Due to NeurIPS submission restrictions, we are unable to include visualizations of our results. Instead, we provide the loss vs. epoch metrics for the text-to-video (T2V) experiments. Specifically, we report the training loss across epochs for GPDiT-H (2B parameters).
> The steadily decreasing loss indicates the potential of our model to generalize effectively to T2V tasks. The detailed results are presented in the table below. Due to current limitations in computational resources and time, we are unable to perform full-scale experiments at this stage.
>
> #### Table 1: Text-to-Video Generation with GPDiT-H on the ConceptualCaption12m and Panda7m Dataset
>
> | Iteration | 500 | 1000 | 1500 | 2000 | 2500 | 3000|
> |-----------|-----|-----|-----|-----|-----|-----|
> | Loss      | 0.168 | 0.142 | 0.142 | 0.138 |0.136| 0.139|
>
>
>
>
> > ### W2: Lack of ablation study
>
> To evaluate the effectiveness of our proposed rotation-based time injection strategy, we conduct an ablation study comparing it with standard AdaLN-Zero and AdaLN-Single [1]. The following results demonstrate that the rotation-based strategy achieves better memory efficiency while maintaining comparable performance to the baseline methods.
>
>
> #### Table 2: Ablation of time-conditioning mechanism  with GPDiT-B on ConceptualCaption12m and Panda7m Dataset
> | Method   | Metric | 10K  | 15K  | 20K  | 25K  |
> |----------|--------|------|------|------|------|
> | Rotation | Loss      | 0.194 | 0.184 | 0.184 | 0.183 |
> |          | Memory | 34G | 34G | 34G | 34G |
> |               | MaxGradNorm | 0.313 | 0.281 | 0.259 | 0.259 |
> | adaLN-Zero    | Loss   | 0.178 | 0.173 | 0.176 | 0.175 |
> |               | Memory | 36G | 36G | 36G | 36G |
> |               | MaxGradNorm | 0.229 | 0.219 | 0.216 | 0.217 |
> | adaLN-single  | Loss   | 0.185 | 0.182 | 0.177 | 0.174 |
> |               | Memory | 34.6G | 34.6G | 34.6G | 3.3G |
> |               | MaxGradNorm | 0.291 | 0.281 | 0.276 | 0.267 |
>
> [1]Pixart-α: Fast training of diffusion transformer for
> photorealistic text-to-image synthesis. arXiv preprint arXiv:2310.00426, 2023.
>
> > ### W3: Empirical validation of effectiveness
>
> We conducted experiments on the GPDiT-B model to evaluate the inference efficiency of our proposed method. To evaluate inference efficiency, we configured the denoising process with 30 steps and generated videos with 33 frames. The following table reports the average time required to generate a single video under this setting.
>
> #### Table 1: Empirical validation of effectiveness on GPDiT-B
> | Method              | inference time (second/iter)|
> | ------------------ | --------------------- |
> | O(F) |     3.34             |
> | O(F^2)| 3.68   |
>
>
>
> > ### W4: Error accumulation
>
> We acknowledge that error accumulation remains a central and unsolved challenge in video generation. By employing teacher forcing and conditioning on clean frames, we suggest our method can alleviate error accumulation in long-range video generation. However, the central contribution of our paper lies in **combining autoregressive and diffusion models in the continuous space**, a direction that has been not fully explored. This unified framework enables us to better model temporal dependencies while maintaining high-quality generation.

---

> > ### Comment · Reviewer_g1aP · 2025-08-06
> >
> > Thanks for the insightful responses. My concerns have been addressed. I keep my rating.

---

> > > ### Author Response · Authors · 2025-08-07
> > >
> > > Thank you again for your valuable feedback. We will continue improving our paper.

---

### Note · Authors · 2025-08-12

We are grateful to the Area Chairs and all reviewers for their thoughtful reviews and valuable suggestions during the rebuttal process.

Our work introduces a unified AR–diffusion framework in a continuous latent space for  video generation, together with lightweight causal attention  and a parameter-free, rotation-based time conditioning that replaces AdaLN-Zero–style conditioning. The framework delivers strong video generation, representation, and few-shot capabilities and naturally extends toward T2V.

Reviewers *xKDW, kYES*, and *FKsG* consistently praise GPDiT as a unified framework for continuous video modeling that integrates continuous autoregressive diffusion with framewise causal attention to enhance long-range temporal consistency. All reviewers emphasize the efficiency contributions, including a lightweight causal attention design, and the parameter-free, rotation-based time conditioning mechanism. Empirically, they note extensive and diverse experiments covering video generation, representation learning, and few-shot learning, which together demonstrate cross-task generalization and multimodal extensibility.

During the rebuttal, we addressed the following reviewer concerns: (i) **Text-to-video comparisons**—we clarified that our model is trained from scratch for video-to-video and added text-to-video experiments with steadily decreasing loss, showing that the modeling choice transfers quickly to new tasks; (ii) **Lightweight model design**—we evaluated our rotation-based time conditioning together with lightweight causal attention and provided ablations against AdaLN Zero and AdaLN Single, confirming comparable or better behavior with lower complexity, alongside efficiency measurements; (iii) **Input context length**—we added studies showing that moderate context improves quality while very long context saturates or degrades performance; and (iv) **Few-shot ability**—**we thank Reviewer *kYES* for suggesting evaluation on unseen tasks**; We trained on real-face-to-ten-style mappings and tested on two unseen styles (using the corresponding style conditions at inference), and the **model produced outputs consistent with the unseen styles, underscoring its generalization to tasks not seen during training and its in-context learning potential.**

All concerns raised by the reviewers have been resolved. We hope these responses assist the reviewers and the Area Chair in better understanding the paper.

---

### Decision · Program_Chairs · 2025-09-17

**Decision:**

Accept (poster)

**Comment:**

The paper introduces a network architecture for diffusion-based autoregressive long video synthesis. The design integrates per-frame-block latent diffusion with causal conditioning on previous clean blocks. Autoregressive video generation is a key research problem towards enabling interactive, streaming video generation beyond the limitation of a fixed video duration.

All reviewers have praised the novel architecture designs of causal attention and time conditioning for improved efficiency and quality. Extensive experiments span video generation, representation, and few-shot learning. The paper is well written with clear contributions beneficial for the broader community.